# Application of L-Cysteine Hydrochloride Delays the Ripening of Harvested Tomato Fruit

**DOI:** 10.3390/foods13060841

**Published:** 2024-03-09

**Authors:** Yunbo Song, Hanzhi Liang, Jiechun Peng, Shenghua Ding, Xuewu Duan, Yang Shan

**Affiliations:** 1Longping Branch, College of Biology, Hunan University, Changsha 410125, China; ybsong0602@163.com (Y.S.); shhding@hotmail.com (S.D.); 2Hunan Provincial Key Laboratory for Fruits and Vegetables Storage Processing and Quality Safety, Hunan Agricultural Product Processing Institute, Hunan Academy of Agricultural Sciences, Changsha 410125, China; 3South China Botanical Garden, Chinese Academy of Sciences, Guangzhou 510650, China; lianghanzhi@scbg.ac.cn (H.L.); pengjiechun@scbg.ac.cn (J.P.); xwduan@scbg.ac.cn (X.D.)

**Keywords:** tomato, fruit ripening, L-cysteine hydrochloride, redox regulation, transcription regulation

## Abstract

Fruit ripening is controlled by internal factors such as hormones and genetic regulators, as well as external environmental factors. However, the impact of redox regulation on fruit ripening remains elusive. Here, we explored the effects of L-cysteine hydrochloride (LCH), an antioxidant, on tomato fruit ripening and elucidated the underlying mechanism. The application of LCH effectively delayed tomato fruit ripening, leading to the suppression of carotenoid and lycopene biosynthesis and chlorophyll degradation, and a delayed respiration peak. Moreover, LCH-treated fruit exhibited reduced hydrogen peroxide (H_2_O_2_) accumulation and increased activities of superoxide dismutase (SOD), catalase (CAT), and monodehydroascorbate reductase (MDHAR), compared with control fruit. Furthermore, transcriptome analysis revealed that a substantial number of genes related to ethylene biosynthesis (*ACS2*, *ACS4*, *ACO1*, *ACO3*), carotenoid biosynthesis (*PSY*, *PDS*, *ZDS*, *CRTISO*), cell wall degradation (*PG1/2*, *PL*, *TBG4*, *XTH4*), and ripening-related regulators (*RIN*, *NOR*, *AP2a*, *DML2*) were downregulated by LCH, resulting in delayed ripening. These findings suggest that the application of LCH delays the ripening of harvested tomato fruit by modulating the redox balance and suppressing the expression of ripening-related genes.

## 1. Introduction

Fleshy fruits play a crucial role in human diets as a primary food source. Ripening is a distinct phase in the life cycle of fruits, involving substantial changes in color, texture, flavor, and nutrition [1]. This process is vital for achieving high-quality commercial fruit. However, over-ripening or senescence is undesirable, as it can reduce marketability and diminish disease resistance [2]. Therefore, gaining a deeper understanding of fruit ripening and senescence could aid in developing methods to enhance sensory attributes and minimize loss during post-harvest handling.

The tomato is an ideal model fruit for investigating the mechanisms that control ripening. Ethylene is widely known as the main initiator for tomato ripening. This plant hormone is produced by two crucial enzymes, ACC synthase (ACS) and ACC oxidase (ACO), and triggers the expression of ripening-related genes through various signal transduction pathways [3]. The intricate regulatory network has been well understood. Key transcription factors like RIN and NOR have been identified as major regulators that orchestrate fruit ripening alongside ethylene signaling [1]. Additionally, epigenetic modifications play a crucial role in the ripening process. In tomatoes, the repression or loss of function of the DNA demethylase DML2 results in DNA hypermethylation and inhibits ripening [4,5]. Recent studies have unveiled the participation of Jumonji C (JmjC)-domain-containing proteins (JMJs), specifically histone demethylases, in the process of fruit ripening. SlJMJ6 acts as an H3K27me3 demethylase that promotes tomato fruit ripening by eliminating the inhibitory H3K37me3 mark of ripening-related genes like *RIN*, *ACS*, *ACO1*, *PL*, and *TBG* [6]. Conversely, SlJMJ7 serves as a negative regulator of fruit ripening in tomato via the direct removal of H3K4me3 from multiple key ripening-related factors and crosstalk between DNA demethylation and histone demethylation [7]. These findings highlight the intricate interplay of internal factors and external signals in governing fruit ripening.

Reactive oxygen species (ROS) are generated during aerobic respiration and act as essential signaling molecules that regulate various physiological processes. However, the excessive accumulation of ROS can cause oxidative damage to macromolecules, hastening the senescence process [8]. Enzymatic antioxidant systems and low-molecular-mass antioxidants help control ROS levels [9,10]. Following harvest, most fruits undergo intense respiration, resulting in ROS accumulation. As fruit senescence proceeds, the balance of redox may be disrupted, causing oxidative stress within the cells. This can result in lipid peroxidation and protein damage, further accelerating the senescence process of the fruit [2]. Studies have shown that the application of antioxidants can eliminate free radicals in the fruit and reduce oxidative stress, thus delaying the senescence and deterioration of the horticultural products, such as litchi, longans, blueberries, or mushrooms [11,12,13,14,15]. Therefore, it is vital to maintain cellular redox equilibrium for slowing the senescence process in fruit. Although antioxidants play an important role in delaying fruit senescence, their regulatory role in fruit ripening and the underlying mechanisms remain unclear.

L-cysteine, a highly potent anti-browning agent, is extensively employed in food industry [16]. However, its stability is compromised in neutral and weakly alkaline solutions, leading to oxidation to cystine. To enhance stability, L-cysteine is often prepared in hydrochloride form. L-cysteine hydrochloride (LCH) is commonly used in food flavorings due to its safety for target species, consumers, and the environment [17,18]. The aim of this study was to examine the effect of LCH on ripening and antioxidant enzyme activities in harvested tomato fruit. Furthermore, we performed transcriptome analyses to uncover the underlying mechanism by which LCH regulates fruit ripening. These findings will contribute to a better understanding of the role of redox regulation in fruit ripening.

## 2. Materials and Methods

### 2.1. Plant Material and Treatments

Tomato (*Solanum lycopersicum* cv. Ailsa-Craig) fruit at the mature green stage were used for this study. The tomato fruit were immersed in solutions of 0.1% LCH for 3 min. As a control, the fruit were immersed in water. Following air-drying for 3 h, the fruit were placed into unsealed polyethylene bags with a thickness of 0.02 mm and stored at room temperature (25 ± 1 °C). Each polyethylene bag contained twelve fruit, and there were eighteen bags in each group. Fruit were sampled at intervals of 0, 5, 7, 9, 11, and 13 days during storage for quality assessment and subsequent analyses.

### 2.2. Fruit Ripening Characteristics

The levels of chlorophyll, carotenoids, and lycopene were quantified through spectrophotometry, as previously described [7], expressed as μg g^−1^ fresh weight. The respiration rate was assessed using a Li-6262 CO_2_/H_2_O analyzer (Li-Cor, Inc., Lincoln, NE, USA) and expressed as mg CO_2_ h^−1^ kg^−1^ fresh weight.

### 2.3. H_2_O_2_ Content and Activities of Antioxidant Enzymes

The H_2_O_2_ content was quantified using a hydrogen peroxide assay kit (Suzhou Comin Biotechnology Co., Ltd., Suzhou, China). The activities of catalase (CAT), ascorbate peroxidase (APX), glutathione peroxidase (GPX), glutathione reductase (GR), and monodehydroascorbate reductase (MDHAR) were determined by monitoring the substrate decrease in the reaction systems at 240 nm, 290 nm, 340 nm, 340 nm, and 340 nm, respectively, as described previously [19], and expressed as nM min^−1^ g^−1^ FW. The activity of superoxide dismutase (SOD) was determined by measuring the enzyme’s ability to inhibit the reduction of nitro blue tetrazolium at 560 nm, as previously described [19], expressed as U g^−1^ FW. One unit of enzyme activity was defined as the amount causing a 50% reduction inhibition.

### 2.4. RNA-Seq Analysis

Total RNA was extracted from the fruit immediately after harvest (0 d) and from the fruits treated with LCH at 7 d. Comparative transcriptome analyses were conducted between control fruit at 0 d and 7 d, as well as between control fruit and LCH-treated fruit at 7 d. The mRNA-seq libraries were prepared using the mRNA-Seq Kit (Illumina, San Diego CA, USA) and sequenced using an Illumina Novaseq 6000 (Illumina, San Diego, CA, USA). The clean reads were aligned with International Tomato Annotation Group version 4.0 using the HISAT2 (v.2.0.1) software. The data analysis was conducted as described previously [7].

### 2.5. Real-Time Quantitative PCR (RT-qPCR) Analysis

Total RNA was extracted from pulp tissue using a FastPure Universal Plant Total RNA Isolation Kit (Vazyme, Nanjing, China). A total of 500 nanograms of RNA was used to prepare cDNA for RT-qPCR. The first-strand cDNA was synthesized using the Prime-ScriptTM RT-PCR kit (TaKaRa, Otsu, Japan). Quantitative PCR amplification was conducted using the ABI7500 Real-Time PCR System (Thermo Fisher Scientific, Waltham, MA, USA). The *ACTIN* gene (Solyc03g078400) was used as the internal control for quantitative normalization. Three biological replicates were conducted. The primer sequences used for RT-qPCR analysis can be found in Appendix A.

### 2.6. Statistical Analysis

Each data point is represented as the mean ± standard errors of three biological replicates. Statistical differences are compared to the control and analyzed using Student’s *t*-test (* *p* < 0.05) with SPSS 19.0 (SPSS, Inc., Chicago, IL, USA).

## 3. Results

### 3.1. Effects of LCH Treatment on Ripening and Pigment Metabolism of Tomato Fruit

The prominent feature of tomato fruit ripening is the reddening of the peel. As shown in Figure 1, the control fruit began to turn red at 7 d and reached edible ripeness at 9 d. The application of LCH effectively delayed the process, resulting in the fruit turning red at 9 d.

The turning red of tomato fruit is associated with the synthesis of carotenoids and lycopene, as well as the degradation of chlorophyll. Consistent with the visual changes in the fruit, the application of LCH resulted in a delay in the pigment metabolism process. Specifically, at 9 d, the levels of carotenoids, lycopene, and chlorophyll in control fruits were 20.7, 17.6, and 3.2 mg/g FW, respectively, while in LCH-treated fruits, these levels were 3.2, 2.8, and 26.5 mg/g FW, respectively (Figure 2A–C).

The tomato is a typical climacteric fruit characterized by a respiration climacteric phase after harvest. As shown in Figure 2D, the respiration rate of the control fruit rapidly increased after harvest, peaked at 7 d, and then declined. LCH treatment delayed the onset of the respiration peak by 2 days and reduced respiration levels afterwards.

### 3.2. H_2_O_2_ Content and Activities of Antioxidant Enzymes

Hydrogen peroxide is a crucial reactive oxygen species produced by organisms and is the main cause of oxidative damage to biological macromolecules. As shown in Figure 3, the H_2_O_2_ content in control fruit increased rapidly at 9 d. Treatment with hydrogen peroxide resulted in a rapid increase in H_2_O_2_ content in fruits after 7 d of storage, while treatment with LCH delayed the accumulation of hydrogen peroxide in fruits.

CAT, APX, GPX, and SOD are essential antioxidant enzymes responsible for eliminating H_2_O_2_, while GR and MDHAR play crucial roles in the regeneration of glutathione and ascorbic acid. In control fruit, the activities of CAT and MDHAR remained stable or decreased slightly at the early stage of storage and rapidly declined at the late stage, whereas the activity of CAT and MDHAR only slightly decreased and was well maintained in fruits treated with LCH (Figure 4). The activities of GPX, SOD, and GR in control fruit increased initially during storage but decreased as storage progressed. In fruits treated with LCH, the activity of SOD was well maintained during later storage, and the activities of GPX and GR were higher compared to control fruit at 13 d (Figure 4). In contrast to the activities of CAT, GPX SOD, GR, and MDHAR, APX activity significantly increased at the early stage of storage and remained at a high level. Surprisingly, the activity was lower in fruits treated with LCH (Figure 4). In summary, the activities of CAT, SOD, and MDHAR in tomato fruit were better maintained via LCH treatment.

### 3.3. Transcriptome Profiling of Tomato Fruit in Response to LCH

To order to unveil the regulatory mechanism influenced by LCH during fruit ripening, RNA-seq analysis was conducted on the control fruit samples at 0 d and 7 d, as well as LCH-treated fruit samples at 7 d. Subsequently, comparative transcriptome analyses were performed between control fruit at 0 d and 7 d, as well as between control fruit and LCH-treated fruit at 7 d. We set the fold-change threshold and false discovery rate threshold for differentially expressed genes at 2 and 0.05, respectively.

A total of 7757 genes, with 3350 upregulated and 4407 downregulated genes, exhibited significant expression differences between the control fruit at 7 d (BR stage) and 0 d (MG stage) (Figure 5A), denoted as ripening-associated genes. Also, a total of 6947 genes (4219 upregulated and 2728 downregulated genes) were identified between the fruit treated with LCH and control fruit at 7 d (Figure 5B), designated as LCH-regulated genes.

Among the identified genes, 5070 were found to overlap between ripening-associated genes and LCH-regulated genes (Figure 5C), with 2953 genes upregulated and 2117 genes downregulated through LCH treatment. DAVID enrichment analysis revealed that the upregulated genes were associated with functions such as plasma membrane, protein kinase, microtubule binding, cell division, photosynthesis, and cell wall organization (Figure 5D). Conversely, the downregulated genes were highly enriched in oxidoreductase activity, fruit ripening, ethylene biosynthesis, ethylene-activated signaling pathways, carotenoid biosynthesis, vitamin C, and signal transduction (Figure 5E).

Furthermore, a substantial number of genes associated with ethylene biosynthesis and response, carotenoid biosynthesis, cell wall degradation, redox regulation, and ripening-related regulators were downregulated by LCH (Figure 6). The accuracy of the RNA-seq data was further validated via RT-qPCR assays. Specifically, the expression levels of 12 fruit ripening-related genes including RIPENING INHIBITOR (*RIN*), NOR-RIPENING (*NOR*), COLORLESS NON-RIPENING (*CNR*), DEMETER-LIKE 2 *(DML2*), 1-aminocyclopropane-1-carboxylic acid synthase 2 (*ACS2*), 1-aminocyclopropane-1-carboxylic acid synthase 4 (*ACS4*), 1-aminocyclopropane-1-carboxylic acid oxidase 1 (*ACO1*), ζ-carotene desaturase (*ZDS*), thioredoxin h1 (*Trxh1*), thioredoxin h17 (*Trxh17*), thioredoxin-like, and methionine sulfoxide reductase B1 (*MsrB1*), were confirmed to be downregulated by LCH (Figure 7). Overall, these findings demonstrate that LCH delays fruit ripening by modulating the expression of genes involved in ethylene biosynthesis and response, carotenoid biosynthesis, cell wall degradation, and transcriptional regulation.

## 4. Discussion

### 4.1. LCH Treatment Delays the Ripening of Harvested Tomato Fruit

Fleshy fruits are commonly categorized into two groups: climacteric and non-climacteric fruits. Climacteric fruits undergo a ripening process and become edible after harvest and then progress to senescence. In contrast, non-climacteric fruits initiate senescence immediately after harvest, resulting in quality deterioration and reduced disease resistance. The regulation of redox status plays an important role in controlling fruit senescence [20]. By modulating the oxidation–reduction balance and managing cellular oxidative stress levels, redox regulation influences senescence and quality in fruits. Studies have shown that antioxidants play important roles in delaying senescence in fruit and vegetables, such as litchi [13,15], longans [12], blueberries [14], mushrooms [11], by scavenging free radicals and reducing oxidative stress damage. Nevertheless, the role of antioxidants in regulating climacteric fruit ripening remains unclear. In this study, the application of LCH, an antioxidant, notably delayed the ripening process of harvested tomato fruit, evident in the delayed color change to red. Furthermore, LCH treatment significantly inhibited the biosynthesis of carotenoids and lycopene, essential for nutritional quality, as well as the degradation of chlorophyll, and climacteric respiration. In addition to controlling the senescence of harvested fruit, our findings suggest that LCH play an important role in regulating the ripening of climacteric fruits. The application of 1-methylcyclopropene (1-MCP), an inhibitor of ethylene perception, has been proven to efficiently delay the ripening process in climacteric fruits [21]. Although cysteine hydrochloride is not as effective as 1-MCP in delaying tomato fruit ripening, it is easier and less costly to use. And as a food additive, cysteine hydrochloride is friendly and safe to humans and the environment [17,18]. Therefore, it provides a simple and alternative approach to extending the shelf life of harvest tomatoes.

### 4.2. LCH Treatment Influences Redox Balance of Harvested Tomato Fruit

ROS are natural by-products of aerobic metabolism in organisms. However, excessive ROS accumulation resulting from the imbalance between production and elimination can lead to oxidative damage to macromolecules, thereby accelerating senescence [8]. Recent studies have highlighted the close relationship between fruit senescence and ROS accumulation, as well as oxidative damage to proteins [2]. Jiang et al. reported that H_2_O_2_ accumulation and protein oxidation intensify as senescence proceeds in litchi fruit [20]. Wu et al. discovered that elevated oxygen levels led to increased H_2_O_2_ accumulation and accelerated senescence in longan fruit during storage, whereas exposure to low oxygen had the opposite effect [22]. Yan et al. [23] reported that treatment with high oxygen concentration resulted in increased H_2_O_2_ accumulation, more severe oxidation damage to proteins and lipids, and accelerated ripening of harvested banana fruit. In this study, we observed a rapid accumulation of H_2_O_2_ in tomato fruit during the later stage of storage, consistent with fruit senescence. LCH treatment reduced H_2_O_2_ accumulation during the later storage period, suggesting that the delayed ripening effect of LCH may be linked to reduced ROS accumulation.

Plants have developed a sophisticated network of enzymatic and non-enzymatic antioxidant mechanisms to counteract oxidative stress [10]. Key enzymatic antioxidants include SOD, CAT, APX, and GPX. SOD facilitates the conversion of superoxide radicals into H_2_O_2_. CAT directly eliminates H_2_O_2_, while APX and GPX break down H_2_O_2_ using ascorbic acid and glutathione as electron donors, respectively [24]. Additionally, GR and MDHAR play crucial roles in the regeneration of glutathione and ascorbic acid, which are beneficial for the activities of GPX and APX [24]. Numerous studies have shown that various postharvest treatments delay fruit senescence and preserve quality by enhancing the activities of antioxidant enzymes, reducing ROS accumulation, and alleviating oxidative damage [25,26,27,28,29,30]. In this study, the response of different antioxidant enzymes to LCH varied. LCH treatment enhanced the activities of SOD, CAT, and MDHAR during later storage, aligning with the reduced H_2_O_2_ accumulation. Interestingly, the activities of GR and GPX were significantly enhanced by LCH at 13 d. Surprisingly, the application of LCH led to decreased APX activities during storage. Thus, LCH treatment enhanced or maintained the activities of CAT, SOD, and MDHAR, and effectively preserved the redox balance, which likely contributed to the reduced H_2_O_2_ accumulation and delayed ripening in harvested tomato fruit.

### 4.3. LCH Treatment Suppresses the Expression of Ripening-Related Genes in Harvested Tomato Fruit

Fruit ripening is a complex and dynamic process characterized by changes in respiration rate, ethylene production, color, texture, flavor, and the transformation of storage substances. This intricate process is orchestrated by a wide array of structural genes and regulatory factors. In this study, we identified a number of ripening-related genes that were downregulated via LCH treatment. These genes are involved in transcription regulation, ethylene synthesis, cell wall degradation, carotenoid biosynthesis, redox regulation, and other ripening-related processes.

Ethylene, a crucial signaling molecule, plays a pivotal role in the ripening of climacteric fruits by initiating ripening processes. Key enzymes like ACC synthase and ACC oxidase are responsible for ethylene production during fruit ripening [31]. Previous studies have shown that tomato fruit ripening is related to the upregulated expression of *ACS2*, *ACS4*, *ACO1*, and *ACO3* [3,32,33,34]. Tao et al. reported that exogenous methyl jasmonate accelerates the ripening of harvested tomato fruit by upregulating the expression of genes linked to ethylene biosynthesis [35]. Conversely, the application of L-Nitro-arginine methylester (L-NAME), a NO synthesis inhibitor, delays the breaker stage of fruits by reducing ethylene release and downregulating the expression of *SlACS2/4* and *SlACO1/3* [36]. Here, we found that *ACS2*, *ACS4*, *ACO1*, and *ACO3* were strongly suppressed via LCH treatment. This suggests that the inhibition of key genes involved in ethylene synthesis is a crucial factor in the delayed ripening of tomato fruit via LCH treatment.

Transcription factors are pivotal in the regulation of fruit ripening. Studies have highlighted the significance of factors like RIN and NOR as key regulators of tomato fruit ripening, influencing the expression of numerous ripening-related genes through both ethylene-dependent and -independent pathways [1,37,38]. Additionally, other transcription factors such as CNR, AP2a, NAC4, and FUL are known to play crucial roles in tomato fruit ripening [1]. Recently, several novel transcription factors have been identified as participating in the regulation of tomato fruit ripening. Jiang et al. demonstrated that SlWOX13, a WUSCHEL-related homeobox transcription factor, plays a positive role in regulating tomato fruit ripening by influencing ethylene synthesis and signaling pathways and by modulating the expression of key ripening-related transcription factors [39]. Fu et al. reported that the transcription factor SlGT31 interacts with the promoters of the ethylene biosynthesis genes *ACO1* and *ACS4* to facilitate the onset of ripening in tomato fruit [40]. Wang et al. discovered that SlHY5 governs fruit ripening at both the transcriptional level by targeting specific molecular pathways and at the translational level by affecting the protein translation process [41]. Our study reveals that LCH negatively regulates key transcription factors like *RIN*, *NOR*, *CNR*, *AP2a*, *NAC4*, and *FUL*, potentially contributing to the inhibition of fruit ripening. DNA methylation serves as a key mechanism for gene expression regulation, which is dynamically controlled by DNA transmethylase and demethylase [42]. Previous studies have demonstrated that DNA demethylase DML2 inhibits DNA demethylation and ripening progression [4,5]. Our study indicates that LCH treatment results in the downregulated expression of *SlDML2*, which might contribute to the downregulation of ripening-related genes and the delay in fruit ripening.

Carotenoid accumulation is a characteristic feature of tomato fruit ripening. Carotenoid biosynthesis is controlled by a multitude of structural genes and regulators [43]. Key genes such as *PSY*, *PDS*, *ZDS*, *ZISO*, and *CRTISO* play a critical role in this process [43]. The regulation of carotenoid accumulation in fruits occurs at multiple levels, including hormonal, transcriptional, post-translational, and epigenetic mechanisms. The role of ethylene in modulating carotenoid biosynthesis entails a complex regulatory network involving RIN, NOR, CNR, AP2a, TAGL1, and FUL1/2 transcription factors [43]. Our findings indicate that treatment with LCH suppresses the expression of these genes (*PSY*, *PDS*, *ZDS*, and *CRTISO*) and upstream regulators (*RIN*, *NOR*, *CNR*, *AP2a*, *TAGL1*, and *FUL1/2*), potentially resulting in a delay in carotenoid and lycopene accumulation in tomatoes.

Fruit softening is an important characteristic of climacteric fruit ripening, primarily driven by modifications in cell wall structure, specifically the solubilization and depolymerization of pectin and hemicellulose [44]. Various crucial hydrolases, such as polygalacturonase (PG), pectin methylesterase (PME), pectate lyase (PL), β-galactosidase (TBG), endotransglycosylase (XTH), and endo-glucanase (CEL), are responsible for cellular wall disassembly during fruit ripening [45,46]. In addition, Expansin (EXP) also plays a role in the regulation of fruit softening [47]. In this study, we observed that treatment with LCH led to the upregulation of cell wall degradation-related genes such as *PG2a*, *PL*, *PL8*, *TBG4*, and *EXP1*, contributing to the delayed ripening of tomato fruit.

## 5. Conclusions

The application of LCH effectively delayed the ripening of harvested tomato fruit, which is related to enhanced antioxidant activity. Furthermore, LCH treatment downregulated a significant number of ripening-related genes involved in ethylene biosynthesis and response, carotenoid biosynthesis, cell wall degradation, redox regulation, and ripening-related regulators. The application of LCH provides a simple and alternative approach to prolonging the shelf life of harvested fruits. However, further research is required to fully elucidate the molecular mechanisms underlying the ability of LCH to retard fruit ripening.

## Figures and Tables

**Figure 1 foods-13-00841-f001:**
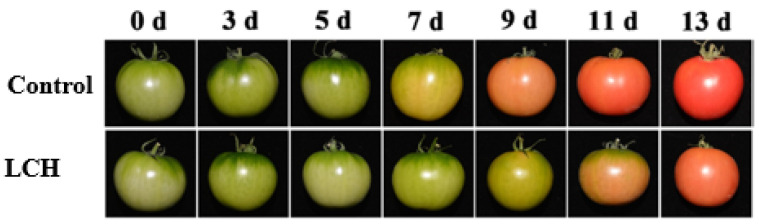
Visual appearance of tomato fruit treated with LCH during storage.

**Figure 2 foods-13-00841-f002:**
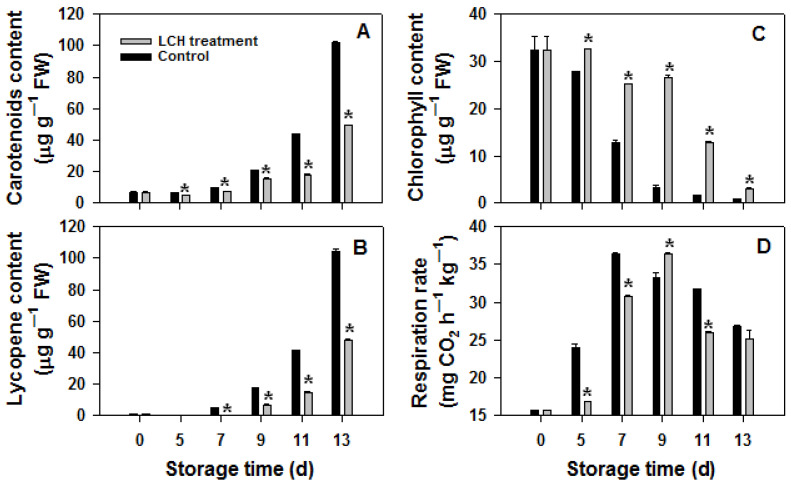
Effects of LCH treatment on the levels of carotenoids (**A**), lycopene (**B**), chlorophyll (**C**), and respiration rate (**D**) in tomato fruit during storage. The data are presented as the mean ± standard error (SE). Asterisks denote statistically significant differences between the control and LCH-treated fruits (Student’s test *p* < 0.05).

**Figure 3 foods-13-00841-f003:**
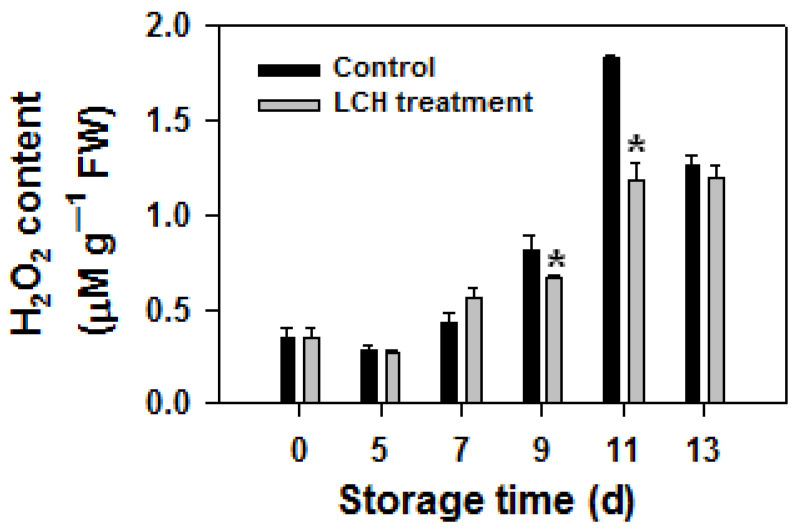
Effect of LCH treatment on H_2_O_2_ content in tomato fruit during storage. The data are presented as the mean ± standard error (SE). Asterisks denote statistically significant differences between the control and LCH-treated fruits (Student’s test *p* < 0.05).

**Figure 4 foods-13-00841-f004:**
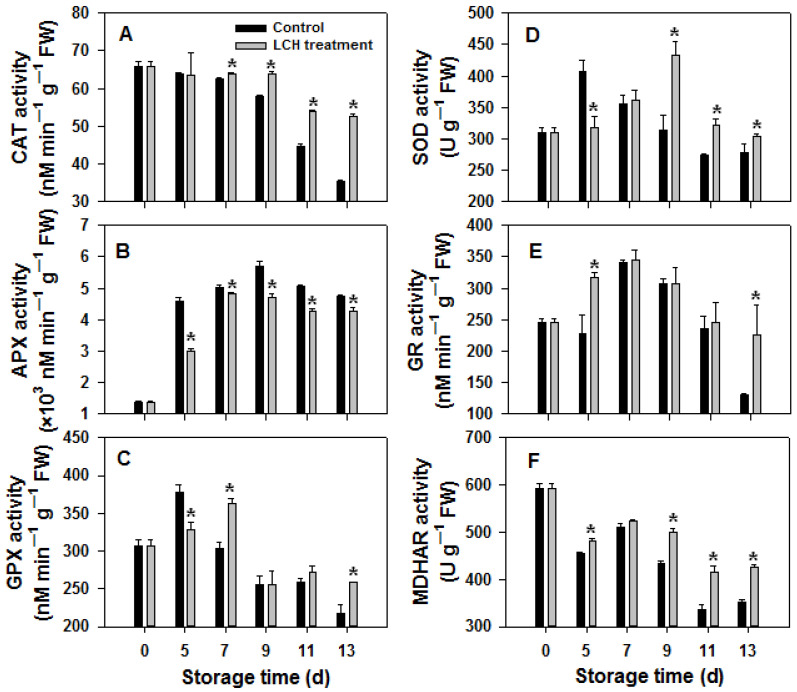
Effect of LCH treatment on activities of CAT (**A**), APX (**B**), GPX (**C**), SOD (**D**), GR (**E**), and MDHAR (**F**) in tomato fruit during storage. The data are presented as the mean ± standard error (SE). Asterisks denote statistically significant differences between the control and LCH-treated fruits (Student’s test *p* < 0.05).

**Figure 5 foods-13-00841-f005:**
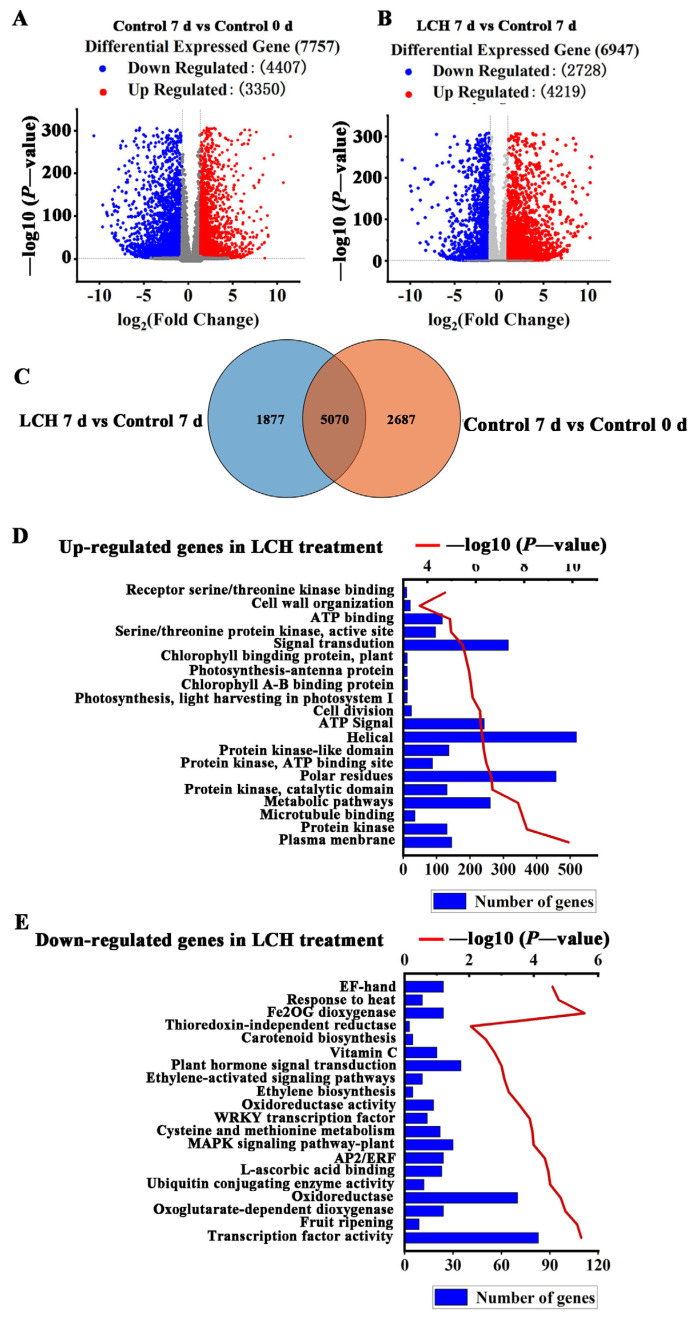
LCH regulates the expression of ripening-related genes in tomato fruit. (**A**,**B**) The number of DEGs identified via RNA-seq. Comparative transcriptome analyses were performed on control fruits at 0 d and 7 d (**A**), as well as on control and LCH-treated fruits at 7 d (**B**). DEGs were defined as genes with a fold-change ratio of ≥2 and a false discovery rate of ≤0.05. (**C**) Venn diagram shows the overlap between ripening-associated genes and LCH-regulated genes. (**D**) DAVID functional clustering analysis of ripening-related genes upregulated by LCH. (**E**) DAVID functional clustering analysis of ripening-related genes downregulated by LCH.

**Figure 6 foods-13-00841-f006:**
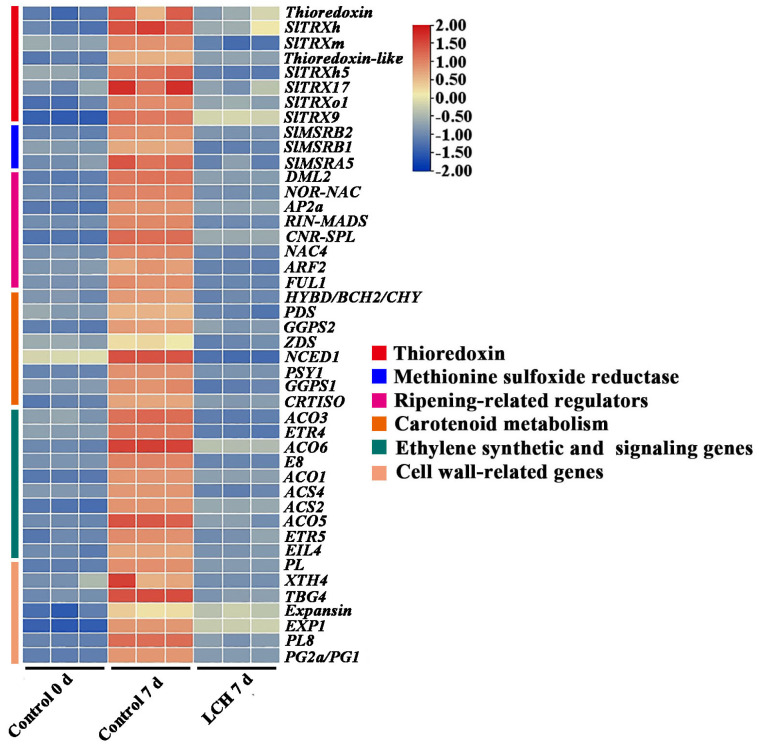
Heat map of the expression levels of key ripening-related genes in control fruit at 0 d and 7 d, as well as in fruit treated with LCH at 7 d.

**Figure 7 foods-13-00841-f007:**
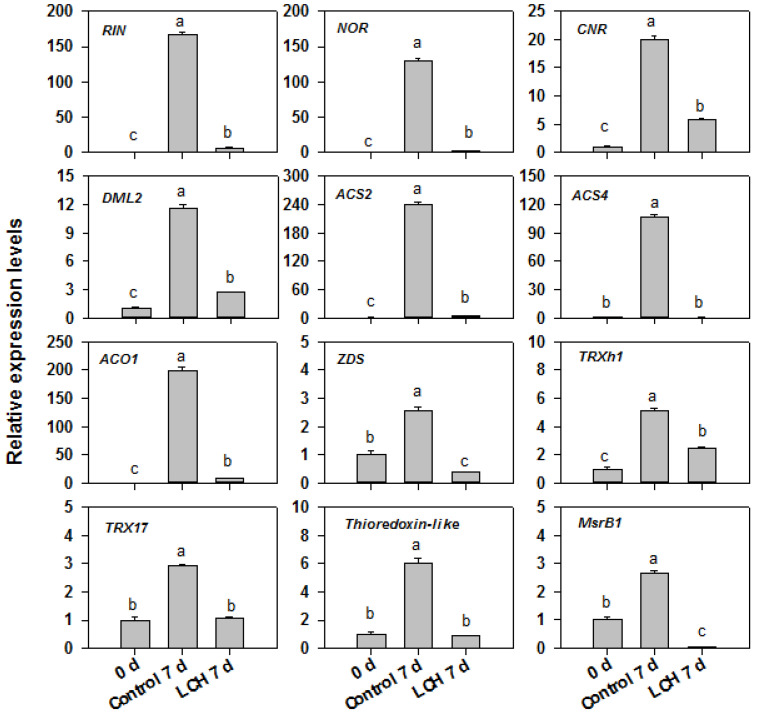
Validation of RNA-seq data using RT-qPCR. Twelve specific genes showing differential expression were validated using RT-qPCR, with *ACTIN* as the internal control. The data are represented as the mean ± standard error (SE). Varied letters above the bars denote statistically significant differences between the samples (Student’s *t*-test, *p* < 0.05).

## Data Availability

The original contributions presented in the study are included in the article, further inquiries can be directed to the corresponding author.

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
