# Peer review of "Application of L-Cysteine Hydrochloride Delays the Ripening of Harvested Tomato Fruit"

_foods, 2024, doi:10.3390/foods13060841_

Round 1

Reviewer 1 Report

Comments and Suggestions for Authors

I have suggested some changes in the attached PDF, which are self-explanatory.

The similarity index should be reduced where it is 3%, 5%, and 6%.

Author Response

Comment: I have suggested some changes in the attached PDF, which are self-explanatory.

Response: As suggested, we have made corresponding revisions.

Comment: The similarity index should be reduced where it is 3%, 5%, and 6%.

Response: We have tried to reduce the repetition rate.

Reviewer 2 Report

Comments and Suggestions for Authors

The manuscript is well-written, scientifically sound, and provides valuable insights into the use of L-cysteine hydrochloride to delay tomato fruit ripening. However, I recommend addressing the following revisions:

The study provides valuable insights into the regulation of fruit ripening, a critical issue for the food industry, particularly concerning post-harvest preservation. However, it narrows its scope to L-cysteine hydrochloride (LCH) without comparing its effectiveness or efficiency with other known or emerging ripening retardants, raising questions about potentially more effective or sustainable alternatives.

While it demonstrates the potential of LCH to delay the ripening of tomatoes, promising to enhance fruit shelf life, the practical applicability in large-scale agriculture and the food industry necessitates considerations of factors like cost, ease of application, food safety, and existing regulations, which are not extensively discussed in the study. These factors are crucial for the real and sustainable implementation of such techniques in the sector.

Moreover, although the use of LCH as a ripening retardant appears to be a promising alternative, the study does not directly address potential environmental or health concerns related to its application, particularly regarding residues in food and the environmental impact of its production and use. These are critical considerations for ensuring long-term acceptance and feasibility of its use in the industry.

Comments on the Quality of English Language

Minor revision

Author Response

Comment: The manuscript is well-written, scientifically sound, and provides valuable insights into the use of L-cysteine hydrochloride to delay tomato fruit ripening. However, I recommend addressing the following revisions:

Response: Thank the reviewer so much for the positive comment.

Comment: The study provides valuable insights into the regulation of fruit ripening, a critical issue for the food industry, particularly concerning post-harvest preservation. However, it narrows its scope to L-cysteine hydrochloride (LCH) without comparing its effectiveness or efficiency with other known or emerging ripening retardants, raising questions about potentially more effective or sustainable alternatives.

Response: Thank the reviewer for the suggestion. We have made corresponding revision in the “4.1 LCH treatment delays the ripening of harvested tomato fruit”

Comment: While it demonstrates the potential of LCH to delay the ripening of tomatoes, promising to enhance fruit shelf life, the practical applicability in large-scale agriculture and the food industry necessitates considerations of factors like cost, ease of application, food safety, and existing regulations, which are not extensively discussed in the study. These factors are crucial for the real and sustainable implementation of such techniques in the sector.

Response: Thank the reviewer for the suggestion. We have made corresponding revision in the “4.1 LCH treatment delays the ripening of harvested tomato fruit”

 Comment: Moreover, although the use of LCH as a ripening retardant appears to be a promising alternative, the study does not directly address potential environmental or health concerns related to its application, particularly regarding residues in food and the environmental impact of its production and use. These are critical considerations for ensuring long-term acceptance and feasibility of its use in the industry.

Response: Thank the reviewer for the suggestion. We have provided some information on the safety of LCH for health and environment in Introduction section.

Reviewer 3 Report

Comments and Suggestions for Authors

The manuscript Foods-2900390 entitled "Application of L-cysteine hydrochloride delays the ripening of harvested tomato fruit” aimed at examining the effects of L-cysteine hydrochloride on ripening, the contents of chlorophyll, carotenoids and lycopene of tomato fruit, and its hydrogen peroxide content, along with the activities of the antioxidant enzymes superoxide dismutase, catalase, ascorbate peroxidase, glutathione peroxidase, glutathione reductase, and monodehydroascorbate reductase. Transcriptome analysis of tomato fruit was performed in response to L-cysteine hydrochloride, towards elaborating on the role of redox regulation in tomato fruit ripening.

In the discussion section the main findings are discussed, namely: the treatment delayed the ripening and influenced the redox balance of the harvested tomato fruit, as well as it suppressed the expression of ripening-rated genes in the harvested tomato fruit, involved in ethylene biosynthesis and response, carotenoid biosynthesis, cell wall degradation, redox regulation, and ripening-related regulators.

The manuscript is well written and in good shape, the provided information is of importance to the topic.

L.50, please kindly provide the explanation of the abbreviation JMJS.

Also, please kindly (i) provide an explanation of why L-cysteine in the hydrochloride form and not the pure one has been chosen, and (ii) commend on its safety for consuming the treated tomatoes after the treatment.

Author Response

Comment: The manuscript Foods-2900390 entitled "Application of L-cysteine hydrochloride delays the ripening of harvested tomato fruit” aimed at examining the effects of L-cysteine hydrochloride on ripening, the contents of chlorophyll, carotenoids and lycopene of tomato fruit, and its hydrogen peroxide content, along with the activities of the antioxidant enzymes superoxide dismutase, catalase, ascorbate peroxidase, glutathione peroxidase, glutathione reductase, and monodehydroascorbate reductase. Transcriptome analysis of tomato fruit was performed in response to L-cysteine hydrochloride, towards elaborating on the role of redox regulation in tomato fruit ripening.

In the discussion section the main findings are discussed, namely: the treatment delayed the ripening and influenced the redox balance of the harvested tomato fruit, as well as it suppressed the expression of ripening-rated genes in the harvested tomato fruit, involved in ethylene biosynthesis and response, carotenoid biosynthesis, cell wall degradation, redox regulation, and ripening-related regulators.

The manuscript is well written and in good shape, the provided information is of importance to the topic.

Response: Thank the reviewer so much for the positive comment.

Comment: L.50, please kindly provide the explanation of the abbreviation JMJS.

Response: We have provided the full name of JMJs in the revised manuscript.

Comment: Also, please kindly (i) provide an explanation of why L-cysteine in the hydrochloride form and not the pure one has been chosen, and (ii) commend on its safety for consuming the treated tomatoes after the treatment.

Response: L-cysteine is unstable in neutral and weakly alkaline solutions, and is easily oxidized to cystine by air. L-cysteine is more stable under acidic conditions, so it is often prepared into hydrochloride form. Therefore, in this study, L-cysteine in the hydrochloride form has been chosen. Cysteine hydrochloride is widely used in food as flavourings. Bampidis et al. (2020a; 2020b) reported that the use of l-cysteine hydrochloride at concentrations up to 25 mg/kg complete feed is safe for the target species, for the consumer and for the environment.

Bampidis, V.; Azimonti, G.; Bastos, M.d.L.; Christensen, H.; Dusemund, B.; Durjava, M.K.; Kouba, M.; Lopez-Alonso, M.; Puente, S.L.; Marcon, F., et al. Safety and efficacy of l-cysteine monohydrochloride monohydrate produced by fermentation using Escherichia coli KCCM 80109 and Escherichia coli KCCM 80197 for all animal species. EFSA J 2020, 18, doi:10.2903/j.efsa.2020.6101.

Bampidis, V.; Azimonti, G.; Bastos, M.d.L.; Christensen, H.; Dusemund, B.; Kouba, M.; Durjava, M.K.; Lopez-Alonso, M.; Puente, S.L.; Marcon, F., et al. Safety and efficacy of L-cysteine hydrochloride monohydrate produced by fermentation using Escherichia coli KCCM 80180 and Escherichia coli KCCM 80181 as a flavouring additive for all animal species. EFSA J 2020, 18, doi:10.2903/j.efsa.2020.6003.